# Bioprospecting the Skin Microbiome: Advances in Therapeutics and Personal Care Products

**DOI:** 10.3390/microorganisms11081899

**Published:** 2023-07-27

**Authors:** Keir Nicholas-Haizelden, Barry Murphy, Michael Hoptroff, Malcolm J. Horsburgh

**Affiliations:** 1Institute of Infection, Veterinary and Ecological Sciences, University of Liverpool, Liverpool L69 3BX, UK; hlknicho@liverpool.ac.uk; 2Unilever Research & Development, Port Sunlight, Wirral CH63 3JW, UK; barry.murphy@unilever.com (B.M.); michael.hoptroff@unilever.com (M.H.)

**Keywords:** skin microbiome, therapeutics, bioprospecting, personal care, ageing, lipid

## Abstract

Bioprospecting is the discovery and exploration of biological diversity found within organisms, genetic elements or produced compounds with prospective commercial or therapeutic applications. The human skin is an ecological niche which harbours a rich and compositional diversity microbiome stemming from the multifactorial interactions between the host and microbiota facilitated by exploitable effector compounds. Advances in the understanding of microbial colonisation mechanisms alongside species and strain interactions have revealed a novel chemical and biological understanding which displays applicative potential. Studies elucidating the organismal interfaces and concomitant understanding of the central processes of skin biology have begun to unravel a potential wealth of molecules which can exploited for their proposed functions. A variety of skin-microbiome-derived compounds display prospective therapeutic applications, ranging from antioncogenic agents relevant in skin cancer therapy to treatment strategies for antimicrobial-resistant bacterial and fungal infections. Considerable opportunities have emerged for the translation to personal care products, such as topical agents to mitigate various skin conditions such as acne and eczema. Adjacent compound developments have focused on cosmetic applications such as reducing skin ageing and its associated changes to skin properties and the microbiome. The skin microbiome contains a wealth of prospective compounds with therapeutic and commercial applications; however, considerable work is required for the translation of in vitro findings to relevant in vivo models to ensure translatability.

## 1. Introduction

Despite perceptions of being an arid wasteland, the human skin is a habitat occupied by a diverse array of microorganisms. Human skin microbiome diversity stems from microenvironment variations between body sites ranging from the *Cutibacterium*-dominated sebaceous regions, rich in lipids, to the halotolerant *Staphylococcus* that are abundant in sweat-rich areas [1,2]. The complement of these microorganisms, the microbiota, utilises host compounds as nutrient sources whilst conveying benefits to their host, such as pathogenic colonisation resistance, local immune system priming and the maintenance of lipid barriers [3]. Akin to macroecological biomes, the skin microbiota community structure consists of organisms specialised to corresponding niches that are susceptible to invasion by foreign species displacing resident organisms [2]. The composition and stability of the skin microbiome are constrained by physical factors that include pH, oxygen concentration, nutrient availability and competition, plus host symbiotic and commensal relationships [4]. Interactions between host microbiota are largely mediated by secreted biological effector compounds, such as digestive enzymes, antimicrobials and quorum sensing compounds [5]. The diversity of skin microbiota and associated effector compounds represent an underappreciated source of effectors potentially relevant to the treatment of a variety of diseases and conditions. 

Bioprospecting is the discovery and exploration of the biological diversity of organisms, genetic components or compounds with potential commercial or therapeutic applications. The utility of beneficial biologically derived compounds is highlighted by the discovery of antibiotics, which has revolutionised modern healthcare and drastically increased global lifespan [6]. Novel drug discovery remains necessary for improving the financial viability of therapeutic and commercial compound pipelines and healthcare outcomes [7]. There is an overwhelming need for novel treatments for bacterial infections in a contemporary world where antimicrobial resistant (AMR) pathogens are an increasing concern, with over 1 million deaths in 2019 associated with such infections [8]. Several alternative treatment strategies are in development to combat AMR infections, such as bacteriophage therapy that has recently gained momentum in Western countries [9].

The skin microbiome provides a vast resource for the development of novel treatment strategies towards the control of AMR infections. Understanding the biochemistry at the interface between host and microbiota will unravel hitherto unknown biology. The coevolution of the human skin with pathogens has resulted in control mechanisms, such as sapienic acid and psoriasin capable of the growth inhibition of AMR *Staphylococcus aureus* and *Escherichia coli*, respectively [8,10,11]. The comparable gut microbiome has revealed a plethora of novel compounds with treatment potential for diseases such as chronic infection, cancer and obesity [12]. Whilst skin microbiota potentially harbours a greater species richness than the gut, its potential abundance of therapeutic compounds and their translational benefits are yet to be mined [3].

Advances in high-throughput genomic technologies have revealed a wealth of microbiome data, facilitating a conceptual shift whereby multicellular organisms’ functions are intrinsically linked to their microbiota [13]. Classically, microbiological investigations relied upon the direct culturing of isolates utilising historic techniques to infer organism identity. The true picture of microbiota diversity was confounded by fastidious organisms lacking the necessary growth conditions provided in vivo, such as complex metabolic interdependences [14,15]. Metagenomic techniques have facilitated more interrogative analyses of the total genomic material present within the skin, which continue to reveal ever more diversity within the skin microbiota [16,17,18,19]. Further development of artificial intelligence technologies has enabled a sophisticated analysis of metagenomic and physical data, allowing for inferences in the causal linkage of skin hydration, age and smoking status to microbiome composition and interpersonal variation [16]. 

Investigations utilising next-generation omics technologies have revealed a multitude of mechanisms and concepts from the skin microbiome. The human skin microbiome is a complex and dynamic field with varied applications, and many organisms and associated compounds have feasibly exploitable properties [3,4,17,18]

This review is distinct through a focus on the breadth and recent advances in the therapeutic and commercial application of biological molecules gained from bioprospecting the human skin microbiome. We further explore the viability of prospective compounds for their translational capacity in human application.

## 2. Skin Environment

The external skin represents a surface functioning primarily to minimise transepidermal evaporative water loss whilst simultaneously providing a physical barrier to extrinsic biological and environmental agents and regulating body temperature [19]. Structurally, skin is comprised of three layers: the innermost hypodermis which stores adipose fat for physical protection and energy, the connective dermis, containing blood and nerve vessels, and the external epidermis [20]. The epidermis is composed of a stratified network of keratinocytes which originate from lower levels and migrate through the epidermis eventually forming the outer stratum corneum of dead anucleate corneocytes [19]. Eventually, corneocytes desquamate and are released from the keratin and lipid matrix, facilitating skin renewal whilst providing nutrition for resident skin microorganisms (Figure 1) [4,21].

Human skin displays localised variability in secretion content, local topography and thickness. Oily skin areas, such as the forehead, contain an increased abundance of sebaceous glands which release sebum onto the skin through the hair follicle canal and holocrine secretions [22,23]. Sebum is composed of a mixture of nonpolar lipids, such as wax esters, squalene and triglycerides, with the latter lipolysed into free fatty acids by resident microbiota [24,25,26,27,28,29,30]. Sebum-derived fatty acids including sapienic acid maintain skin pH and exert a broad-spectrum antimicrobial action modulating skin microbiota composition. Community structure disruption is linked to inflammatory skin conditions like atopic dermatitis [31,32,33]. Epidermal lipids secreted directly by keratinocytes filling in areas between squamous corneocytes act directly to minimise transepidermal evaporative water loss [34]. 

Perspiration occurs across nearly all bodily regions via sweat glands secreting saline-rich liquid onto the skin to facilitate evaporative thermoregulation [35,36]. The regional variation in human skin sites is further evidenced through variations in thickness, invaginations, moisture content and hair follicle density, among other factors (Figure 1), [2].

**Figure 1 microorganisms-11-01899-f001:**
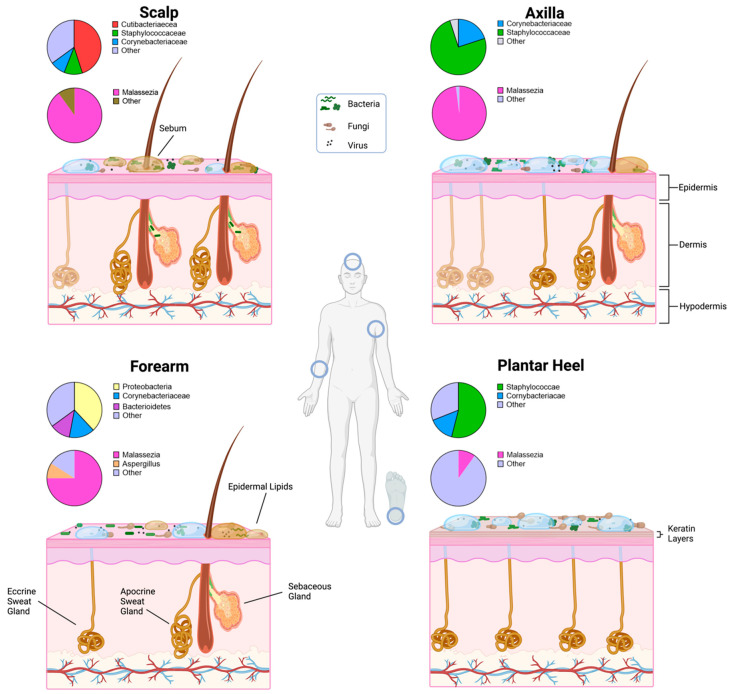
Structure and microbial diversity of healthy human skin at distinct body areas. The microhabitats of the scalp, axilla (armpit), forearm and plantar heel are related to distinct secretion profiles associated with epidermal lipids, sebum and sweat. Lipid- and sebum-rich sites are associated with an increased proportion of lipophilic microorganism genera, namely, *Cutibacterium* and *Malassezia.* Sweat-rich regions display an abundance of halotolerant genera such as *Staphylococcus* and *Corynebacterium* [1,37]. Pie charts represent the abundance of bacterial and fungal genera at corresponding body sites adapted from several studies. Additionally, sites such as the plantar heel display a high degree of diversity across individuals [2,38,39]. Created with www.BioRender.com (accessed on 4 April 2023).

## 3. Drivers of Skin Microbiome Community Structure

The human skin microbiome is governed by fundamental ecological principles relating to habitat, species interaction and community disruption. The skin environment contains an abundance of biological niches that vary in factors, including topography, lipid content, pH, salinity and moisture content alongside temporal changes from ageing and gender [16,40,41]. Human skin is occupied by very diverse populations of bacteria, archaea, fungi, viruses and protists, with colonisation requiring specialisation and adaptation due to niche-specific microenvironments [42,43]. Resident skin microorganisms coevolved with competitors and their hosts, producing complex nutritional interdependencies and behavioural characteristics such as diverse antimicrobial biosynthesis [2,4,44]. Given the range and variation in skin environments alongside differences in external factors, microorganisms are under a variety of abiotic and biotic selective pressures driving variation in stable community structures (Figure 2), [3].

## 4. Pathogenic Invasion and Community Disruption 

The skin microbiome is susceptible to invasion by foreign organisms capable of disrupting the community equilibrium. Many human skin pathogens are generally considered commensal, with their opportunistic pathogenic behaviour evident when introduced into a new subsurface niche through environmental disturbance, such as a laceration [46]. Commensals such as *Staphylococcus epidermidis* and *Cutibacterium acnes* provide host benefit through nutritional competition with pathogens, stimulating host antimicrobial peptide secretion and dampening nonpathogenic local immune system activation [47,48,49]. Yet *S. epidermidis* and *C. acnes* represent common aetiological agents of nosocomial infections of immunocompromised patients, generally following medical device implantation and prosthetic joint surgery [4,50]. *Staphylococcus aureus* asymptomatic colonisation of the anterior nares is often causative of future systemic infection, a threat compounded by the feasibility of AMR strains competing with susceptible strains for dominance [51]. The wide range of skin-relevant pathogens has been discussed in detail elsewhere [52,53,54]. Current antibiotic treatment strategies are often ineffective given the recalcitrant nature of skin-pathogen-associated biofilms alongside the prevalence and continued dissemination of multiple antibiotic resistance elements [55,56,57]. Mechanisms by which microorganisms interact directly influence the microbiota community structure, hence these mechanisms can be manipulated as a source of therapeutically and commercially relevant compounds.

## 5. Therapeutically Relevant Skin-Microbiome-Derived Compounds

### 5.1. Bacteriocins 

Bacteriocins are a range of bacterial-derived peptides and proteins capable of exerting bacteriostatic and bactericidal activity. Bacteriocins act as competitive factors to inhibit the growth of adjacent organisms, akin to antibiotics, yet are associated with distinct resistance mechanisms and thus therapeutically relevant [58]. Diversity in the structure of bacteriocins has evolved with variations in their molecular targets and breadth of host range [59,60,61].

In contrast to broad-spectrum antibiotics, bacteriocin-based treatments could be tailored to the clearance of specific pathogenic organisms with minimal disruption to natural flora [62,63,64,65]. Hence, they are relevant in the treatment of nosocomial infections which are linked to unstable patient microbiomes and increased colonisation by reservoirs of highly antimicrobial-resistant pathogenic bacteria in the healthcare environment [65,66].

Bacteriocins have been identified as effector agents of bacterial antagonism, directing research to exploit their commercial and therapeutic activity [67]. Commercial applications of bacteriocins have focused primarily on the preservation of meat, vegetables and dairy products through the prevention of the bacterial degradation of food [68]. To date, Nisin derived from the dairy fermentative lactic acid bacterium *Lactococcus lactis* remains the sole Food and Drug Administration-approved bacteriocin with licensing in food preservation [69]. The scope for the therapeutic use of bacteriocins is broad, ranging from infection treatments to oral hygiene products and spermicidal compounds. The therapeutic bacteriocin field has yet to fully target the potential of skin microbiome diversity, with studies needed to further the elucidation of the biosynthetic pathways, mechanisms of action and cost of production alongside tolerance in clinical trials [70,71,72].

Studies of the nasal microbiome identified that coagulase-negative commensal staphylococci produce bacteriocins active against niche competitors [73]. The skin and nasopharynx commensal *S. lugdunensis* synthesises lugdunin, which represents the first nonribosomal-produced antimicrobial identified within the human skin microbiome. Classified as a novel class of thiazolidine cyclic peptide bacteriocin termed fibupeptides, lugdunin inhibits the growth of several Gram-positive bacteria, including methicillin-resistant *S. aureus* (MRSA) [74,75,76]. Lugdunin is proposed to exert antimicrobial activity through the disruption of the transmembrane pH gradient likely leading to intracellular protein denaturation and reduced proton motive force, directly inhibiting respiration [74,77,78]. Lugdunin is further evidenced to increase the gene expression of several cutaneous antimicrobial peptides and induce the phagocytic recruitment of neutrophils and monocytes [75]. A key driver of lugdunin research is the low propensity for resistance development in exposed bacteria, with one study identifying *S. aureus* as not developing resistance after several subinhibitory passages [76].

The therapeutic translation of purified bacteriocins remains difficult due to several factors, such as serum half-life and immunogenicity. Protein engineering represents a promising area for redesigning bacteriocins towards their human use. An example comes from a study of the domesticated ruminant commensal *Staphylococcus simulans* that is associated with rare opportunistic infections in humans, such as endocarditis and erythema [79,80,81]. *S. simulans* secretes the bacteriocin lysostaphin, an endopeptidase enzyme capable of pentaglycine bridge cleavage within the peptidoglycan of staphylococci. Lysostaphin exerts anti-*S. aureus* bactericidal activity regardless of planktonic, quiescent and biofilm-associated lifestyles [82]. Lysostaphin immunogenicity represents a major barrier to clinical application. Protein engineering reduced lysostaphin immunogenicity through T-cell epitope removal alongside an increased serum half-life through polyethylene glycol and albumin conjugation [83,84,85]. Various lysostaphin application vehicles have been effective versus *S. aureus* in vivo via intravaginal dissolvable tablets, emulsion-based topical gels and intravenous injection.

The viricidal, virostatic and food-preservative applications of certain bacteriocins have further been reported. For example, the gut commensal and rare skin pathogen *Enterococcus faecium* produces the bacteriocin CRL35, capable of the considerable inhibition of herpes simplex virus (HSV) 1/2 through late-stage protein-synthesis retardation in vitro [86,87,88]. Antimicrobial properties of CRL35 have been demonstrated in vivo and been explored for use as a dairy food preservative via chitosan microencapsulation with a retainment of bactericidal activity [89]. *Bacillus subtilis* can be found on skin and its bacteriocin subtilosin, a macrocyclic lantibiotic, is capable of the inhibition of both HSV-1 and 2 late-stage replication in addition to several bacteria associated with urinary tract infections [90,91,92]. Further nonskin-bacteria-derived bacteriocins have potent antiviral activity, which highlights the scope for further investigation of the skin microbiome in this area [90].

Most skin-relevant bacteriocins remain to be explored in preclinical development stages to assess pharmacological benefits. Several considerations must be addressed prior to the therapeutic and commercial applications of bacteriocins, including toxicity, biological half-life and the capacity for resistance development [58]. Indeed, given the range of susceptibility spectrums for bacteriocins, there is a need to collect temporal data as a means to disfavour those with the potential to disrupt the skin microbiota community, where it may exacerbate or perpetuate inflammatory disorders [67].

### 5.2. Bacteriophages

The virome remains an underappreciated area of the skin microbiome, with attention mostly limited to bacteriophages. In natural environments, bacteriophages are considered to modulate bacterial communities through the proliferation inhibition of dominant species, thereby maintaining diversity via predator–prey coevolution [93]. Research of the species-rich gut virome has revealed 10^10^ bacteriophages per gram of faecal material, with variations indicating complex ecological interactions [94,95]. Skin metagenome analyses have highlighted the breadth of phages associated with the predominant skin bacterial genera *Staphylococcus, Cutibacterium* and *Streptococcus* [96,97]. This repertoire represents a rich source of potentially therapeutic bacteriophages.

A principal translational pathway for the therapeutic use of lytic bacteriophages is to clear pathogens from infected tissue [98]. Phages exert bactericidal activity with typical strain specificity for infection through the interaction with bacterial cell surface structures, such as lipopolysaccharide or teichoic acid [99,100]. The genetic manipulation of phage–bacteria determinant regions has been extensively studied, thereby improving the effectiveness of this strategy for the treatment of AMR infections [101,102,103].

Multiple bacteriophages are reportedly being explored for phage therapy for various manifestations of skin-associated pathogens. Support for clinical research comes from phage therapy success for skin infections of *Mycobacterium chelonae* and multiple-drug resistant (MDR) *Pseudomonas aeruginosa* [104,105]. Topical bacteriophage therapy targeting *C. acnes* has advanced to phase I clinical trials after displaying efficacy in reducing *C. acnes* abundance without significant safety concerns [106]. Screening studies isolated seven distinct bacteriophages effective against *S. epidermidis* albeit with highly variable strain-specific resistance profiles [107]. Combinations of *S. epidermidis* with *S. aureus*-specific phages were demonstrated to suppress phage-resistant mutants of *S. aureus* in a skin model of atopic dermatitis [108].

More than 10 clinical trials were in various stages of completion by early 2023 investigating the efficacy of bacteriophage therapy for the treatment of musculoskeletal, skin and soft tissue infections [109]. Previous trials supported the high safety of phage therapy with varying efficacy arising from delivery mechanisms and phage stability [110]. For example, PhagoBurn is a phage-treated wound dressing found safe yet ineffective in phase I/II clinical trials at reducing *Escherichia coli* and *Pseudomonas aeruginosa* burdens, likely through insufficient dosing. The foundation is established, however, for future phage-mediated skin infection trials using phage lysins, liposome-encapsulated phage cocktails and antibiotic phage combinatorial therapy [111].

A topical bacteriophage cocktail therapy for acne, BX001, demonstrated significant *C. acnes* abundance reduction in phase I cosmetic clinical trials. The evidence implicates the IA-1 *C. acnes* phylotype as an opportunistic pathogen of the pilosebaceous unit by stimulating a localised proinflammatory response leading to acne lesions, meaning such treatment has utility [112]. Direct modulation of skin microbiome composition through targeted phage therapy may yield novel skin treatment strategies from wound healing to eczema [108,113,114,115]. A sufficient knowledge of multiple phages with distinct cellular targets is required to mitigate the capacity for phage resistance alongside a physiochemical knowledge of product formulations slowing product development [116].

An appreciation of safety concerns is necessary for the responsible translation of bacteriophages into clinical use. Wildtype phages are classified in relation to their infection cycles, with temperate phages associated with prophage genomic integration and maintenance, whilst obligate lytic phages are associated with direct cellular lysis following infection [117,118]. Phage therapy focuses on obligate lytic phages to circumvent toxin production, integration and insertional mutagenesis [119]. Temperate phage insertions are experimentally evidenced in some cases to facilitate the mobilisation of pathogenicity islands and associated antibiotic resistance and superantigen genes in *S. aureus*, thereby limiting their translation by promoting virulence and treatment resistance [120,121]. Strategies to circumvent inherent phage limitations include the modification of phage DNA to encode CRISPR-Cas9 systems that resensitise resistant bacteria to antibiotic treatments or bactericidal toxins [122]. Bacteriophage endolysins targeting specific cell wall lysis are a promising alternative to hindrances associated with bacteriophage therapy by removing the replicative cycle of phages, albeit with higher production costs [123]. Endolysins available commercially include the eczema treatment Gladskin Micreobalance^®^ with considerable mitigation of *S. aureus*-associated atopic dermatitis [124].

### 5.3. Cutaneous Lipids

Cutaneous fatty acids impart multiple functions for the host, namely, microbial growth inhibition through a reduction in skin pH alongside direct antimicrobial activity, whilst simultaneously promoting commensal adherence [2]. Skin fatty acids are produced through the hydrolysis of epidermal and sebaceous triglycerides, with activity predominantly facilitated through the extracellular lipase secretion of commensals, such as *Cutibacterium* spp. and *Staphylococcus* spp. [125,126,127]. Fatty acids further function as signalling molecules capable of diminishing proinflammatory cytokine induction in keratinocytes associated with allergic dermis reactions through free fatty acid receptor 1 activation [128]. Triglyceride degradation liberates glycerol, a potent humectant that supports corneocyte desquamation from corneodesosome cleavage, thereby stimulating the recovery of irritated skin [26,129,130]. The microbial fermentation of glycerol with the production of fatty acids, namely lactic acid, promotes an acidic pH and the upregulated gene expression of essential skin barrier proteins in keratinocytes [131]. Topical formulations of certain skin lipids have the potential for therapeutic use due to their range of activity and antimicrobial activity. Sphingosine and sapienic acid show potent activity towards staphylococci [11,31,132]. Skin lipids represent an avenue for therapeutic application, and studies indicate a potential use for sapienic acid as a stable antimicrobial in cosmetic products [31,132,133].

### 5.4. Biofilm Inhibitors

Microorganisms aggregate into biofilms through the secretion of matrix proteins, polysaccharides and DNA, with the structures considered a dominant growth mechanism by members of the skin microbiome [134,135]. Biofilms convey many benefits to their community with decreases in both immune system stimulation and antimicrobial uptake through metabolic dormancy [136]. Many opportunistic skin pathogens form monomicrobial biofilms, e.g., *S. aureus*, *C. acnes* and *S. epidermidis*. Notably, *S. aureus* biofilms are prevalent in atopic dermatitis, with a correlation between abundance and disease severity [137]. *C. acnes* and *S. epidermidis* biofilms are associated with medical-implant-associated infection, often leading to systemic bacteraemia in immunocompromised individuals [138,139,140]. Given the clinical relevance of biofilm-associated infection, novel therapeutics are necessary to circumvent the treatment resistance of these bacterial aggregates.

Resident skin flora exerts antagonistic interactions to limit biofilm formation and integrity through the production of antibiofilm agents. *S. epidermidis* secretes a serine protease, Esp, evidenced to disassemble MRSA biofilms by degrading several proteins for cell wall and biofilm formation. Esp can degrade human receptors utilised by *S. aureus* for adherence, such as fibronectin to convey colonisation resistance [141,142,143]. Moreover, a *S. epidermidis* protease-independent biofilm inhibitor is effective against a range of MRSA and methicillin-susceptible *S. aureus* (MSSA) without reducing cell viability. This uncharacterised biofilm inhibitor was proposed to be a phenol-soluble modulin (PSM) likely functioning through the inhibition of the polysaccharide adhesion operon [144,145,146].

The lipophilic, aerotolerant commensal *Cutibacterium acnes* can reside in pilosebaceous units, and its secreted lipases hydrolyse triglycerides, releasing short-chain fatty acids (SCFAs) resulting in local pH reduction and the inhibition of competing bacteria, stabilising niche occupation [147]. Additional beneficial functions of *C. acnes* include the secretion of antioxidant enzymes and in vivo antitumoral properties [49,148]. However, sebum-derived SCFAs may also induce proinflammatory gene expression in keratinocytes and sebocytes within the pilosebaceous unit that likely drive acne vulgaris manifestations [149,150]. SCFAs further inhibit the polysaccharide biofilm formation of *S. epidermidis* at physiological concentrations whilst simultaneously increasing susceptibility to both ampicillin and doxycycline [151]. SCFAs are being explored within the gut microbiome given their significant potential for inflammatory and immune diseases, with a likely relevance to cutaneous homeostasis [152,153,154].

### 5.5. Quorum Sensing Modulators

The survival of bacterial communities requires constant environmental adaptation to selective pressures [155]. Quorum sensing (QS) systems mediate cell-to-cell communication to bring about a coordinated change in gene expression producing a community beneficial behaviour [156]. QS relies upon the secretion of autoinducer compounds at subthreshold levels that diffuse into the extracellular environment where the autoinducer concentration is linked with bacterial population density. The post-threshold autoinducer activates signal transduction to a transcription factor(s) that drive(s) regulon expression [157,158,159,160].

QS products such as extracellular proteases have a metabolic expense but provide community level fitness benefits, hence collaborative production acts to limit individual cost [161]. Many QS-mediated products are virulence factors that facilitate processes such as biofilm formation, integrity promotion and dispersal associated with systemic disease progression [160,162]. In recent years, analogous quorum-sensing-like systems were identified in fungi and bacteriophages, highlighting the significance of microbial communication in microbiota present on the skin [163,164]. 

The secretion of QS inhibitors represents a form of antagonism common within the skin microbiome that has potential therapeutic translatability. Various commensal staphylococci, including *S. epidermidis*, *S. hominis* and *S. simulans*, secrete inhibitors of the *S. aureus* accessory gene regulator (agr) that interfere with colonisation and virulence [165,166]. *S. hominis* autoinducer peptides (AIPs) were shown to quench accessory gene regulatory (agr) QS systems of MRSA, with AIP-2 providing protection from *S. aureus* necrosis and skin damage [167]. *Staphylococcus warneri* produces AIP-1 and AIP-2 capable of the dose-dependent inhibition of *S. epidermidis* agr-1 and all agr of *S. aureus*, reducing virulence factor production and providing protection against associated skin barrier damage [168].

Since pathogenic bacteria often utilise QS to stimulate biofilm formation, many QS inhibitors have a translational application in the treatment of recalcitrant biofilm infections derived from the human skin [169,170]. The skin commensal *Staphylococcus xylosus* expresses an RNAIII-inhibiting peptide (RIP) that inhibits QS-associated signal transduction pathways. RIP is a potent inhibitor of *S. aureus* and *S. epidermidis* biofilm formation proteins [171,172,173]. Moreover, RIP treatment displayed both an in vitro inhibition of *S. aureus* adhesion and efficacy in the treatment of *S. aureus* murine infection models [173,174].

### 5.6. Fungicidal Compounds

The cutaneous mycobiome is less well studied as the fungal component that colonises human skin. The mycobiome becomes established following birth with compositional changes associated with extrinsic and intrinsic factors comparable with the microbiome as a whole. Skin fungal dysbiosis correlates with common cutaneous disorders, such as dandruff, atopic dermatitis and pityriasis versicolor [175,176,177]. Fungal pathogen emergence is underrecognised in contrast to the antimicrobial resistance of bacterial pathogens and viral pathogen resurgence [178,179]. Problematically, fungal infection treatment is generally limited to just four classes of compounds, which can often be unsuccessful due to resistance development that highlights the necessity of novel antifungal compounds [180,181].

The predominant genus of the human cutaneous mycobiome is *Malassezia*, which was recently identified to have a greater breadth of species from metagenomic studies, albeit species that remain uncultured [182]. *Malassezia* can produce indole compounds for nitrogen acquisition through tryptophan metabolism [183]. *Malassezia* indoles display broad-spectrum fungicidal properties, effective against pathogenic yeasts and moulds at skin-relevant concentrations in vitro [184]. *Malassezia furfur* produces several indoles, such as indolo [2,3] carbazole, which inhibits *Candida* spp. and other *Malassezia* spp. at concentrations less than 6 ug/mL [184,185]. Known ligands of aryl hydrocarbon receptors, these indoles can result in the induction of proinflammatory responses, drastically limiting their therapeutic prospects [186].

Several commensal bacteria inhibit *Candida albicans* colonisation within the cutaneous environment. *S. epidermidis* stimulates the production of specific CD8+ T cells via localised dendritic cell activation, resulting in an improved protection against *Candida albicans* infection through enhanced innate defence systems [187]. *S. epidermidis* stimulates the primary keratinocyte production of human antimicrobial peptides beta defensin 1–3 in a Toll-like receptor-2-associated manner [177,188]. *Lactobacillus* spp. that are abundant within the vaginal microbiome produce lactic acid and SCFA resulting in vaginal acidification that inhibits the *C. albicans* yeast–hyphae transition [189]. Such hyphal transition represents a key virulence strategy of *C. albicans* required for mucosal invasion and systemic dissemination [190]. Several postulated antifungal benefits of vaginal lactobacilli are related to colonisation resistance through strong adhesion and biofilm formation upon mucosal membranes and free-radical secretion increasing mucus cohesion [191]. Indeed, vaginal probiotics are commercialised with *Lactobacillus*-based Canesten Canesflor^®^, demonstrating the potential of developing probiotics for fungal infections. The antifungal potential of *Saccharomyces cerevisiae* and *Candida* spp. was demonstrated in vitro to inhibit biofilm formation and hyphal transition in *C. albicans*, highlighting the need for future probiotic research [192]. 

### 5.7. Skin Cancer Treatments

The development of novel cancer treatments represents an unequivocally imperative task for modern health science. The significance of cancer is highlighted through the rapid growth in global morbidity and mortality, representing a leading cause of premature deaths in an estimated 60% of countries as of 2021 [193,194].

Within cancerous tumours exists a distinct microenvironment, and the microbiome can alter both progression and treatment resistance. Multiple associations were identified between the prevalence of individual skin microbiome members and specific skin cancers, such as *S. aureus* within squamous cell carcinoma skin biopsies [195]. A causative relationship between skin microbiome members and skin cancers is proposed to occur through the induction of proinflammatory responses, which links to an apparent increased risk of skin cancer with skin-microbiome-associated disorders [196,197,198,199]. There is a necessity to investigate the intentional modification of skin tumour microbiomes to mitigate the risks associated with skin cancer development given the previously described rich source of bactericidal products [195]. The characterisation of the skin microbiome has so far revealed several compounds displaying potential antioncogenic activity capable of increasing the available therapies to clinical professionals.

The metabolic analysis of *S. epidermidis* skin isolates revealed the trait variable production of 6-N-hydroxyaminopurine (6-HAP), a hydroxylamine adenine analogue with mutagenic and teratogenic activity [200,201]. 6-HAP displayed an in vitro selective inhibition of lymphoma and melanoma tumour cell line proliferation whilst exerting relatively little effect on an epidermal keratinocyte cell line. *S. epidermidis*-producing 6-HAP is prevalent in the human skin microbiome, and these strains exerted a significant inhibition of ultraviolet-associated neoplasia generation in vivo [202]. The antioncogenic properties of 6-HAP have been questioned based on the characterisation of commercially sourced versus commensal-produced 6-HAP [203,204]. The therapeutic potential of 6-HAP is contingent on the further translation to in vivo models alongside the elucidation of its mechanism of action, prior to its application in cancer treatment and prevention.

Commensal *Malassezia* spp. synthesise indirubin, a compound capable of potent aryl hydrocarbon receptor (AHR) agonism, with a proposed competitive inhibition of cyclin-dependent kinases associated with cell cycle modulation, resulting in cell cycle arrest [205]. Further antioncogenic mechanisms of indirubin have been discussed in detail elsewhere [206]. Despite the apparent commensality of *Malassezia* spp., opportunistic pathogenesis is associated with several conditions, such as dermatitis, supporting the application of topical probiotic formulations, i.e., for treatment of skin cancers [207].

Bacteriocins represent an attractive source of antioncogenic compounds arising from their specificity, nontoxicity and abundance within the skin microbiome [208]. Pyocins (bacteriocins produced by the genus *Pseudomonas*) of the transient opportunistic pathogen *Pseudomonas aeruginosa* selectively inhibit the proliferation of human hepatoma, B lymphocyte and murine fibroblast cell lines [209,210]. The antioncogenic activity of bacteriocins produced by skin-relevant pathogens has been described, such as *Klebsiella pneumoniae*-derived microcin E492; however, the therapeutic potential of skin-microbiome-derived bacteriocins remains unexplored [208]. Moreover, investigations of the antioncogenic activity of bacteriocins are largely limited to in vitro cell lines, illustrating the necessity for in vivo studies and considerations of the reported limitations of the therapeutic utilisation of bacteriocins.

Many anticancer therapeutic compounds are limited by the development of resistance in target cells alongside nonselective cell targeting, resulting in multiple adverse side effects. The selective and distinct functional mechanisms of bacteriocins highlight their therapeutic application potential [209]. Several methods were discussed to feasibly increase bacteriocin therapeutic activity, for example, the conjugation to nanoparticles for synergistic drug delivery facilitating a lower therapeutic dosage [211]. Additionally, many common oral probiotic compounds are associated with the production of bacteriocins with considerable anticancer activity, highlighting the need for parallel investigations with topical probiotics or purified compounds [209].

## 6. Skin Microbiome Applications in Personal Care Products

Personal care products represent a considerable proportion of the global economy, with a total estimated market valuation of over USD 500 billion in 2022 and a predicted annual growth rate of over 6% [212]. Within this growth potential, the skin microbiome has many demonstratable applications for personal care product development and many areas for future investigations.

The composition and stability of an individual’s skin microbiota is paramount for the maintenance of cutaneous homeostasis through functions such as pathogenic colonisation resistance [2]. A growing body of evidence correlates skin microbiota composition with biophysical skin properties alongside the perpetuation and onset of skin conditions [16,40,213,214,215,216,217]. Hence, the reshaping of the skin microbiota community structure and utilisation of isolated effector compounds represent a significant area for commercial product development [218]. The viability of such products is shown by the prevalence of novel topical probiotics in the personal care market alongside the renowned success of parallel gut probiotics.

### 6.1. Acne Vulgaris

The key microbial component of acne pathophysiology is linked to the obstruction of the sebaceous gland leading to sebum accumulation, resulting in *C. acnes* extracellular lipases creating proinflammatory SCFAs [219,220,221]. Several clinical trials have demonstrated probiotics cause *C. acnes* inhibition leading to reduced acne severity. A topical mixture of *E. faecalis* and *Lactobacillus* sp. applied bidaily resulted in reductions in acne lesions, proposedly resultant of *C. acnes* inhibition and reduced proinflammatory factor production. Similarly, trials utilising aqueous *Lactobacillus plantarum* probiotics demonstrated dosage-dependent reductions in lesion size [222]. To date, these trials have had a small cohort of less than 100 people, inviting larger studies to support the confidence and reliability of such products [223,224]. Skin probiotics capable of reducing acne manifestations include products recently released to the market, such as those from Yun probiotherapy, indicating the growth of such probiotic markets [225].

### 6.2. Atopic Dermatitis

Atopic dermatitis (AD) is the most common inflammatory skin disorder globally and is associated with pruritic and desiccated inflamed lesions, arising from various inheritable mutations in skin barrier proteins and allergen-specific immunological defects [226,227]. AD cutaneous dysbiosis is characterised by an increased abundance of staphylococci such as *S. aureus* and *Malassezia* spp. which perpetuates AD manifestation and its severity through the secretion of virulence factors (e.g., toxins) and cytokine-inducing nanovesicles, respectively [228,229,230]. Nakatsuji et al. identified that in a small cohort study, the transplantation of specific antimicrobial peptide-producing *S. epidermidis* and *S. hominis* strains yielded considerable reductions in *S. aureus* abundance in AD patients [228]. Such promise means further investigations with a larger cohort size and an understanding of the longer-term persistence of transplanted strains with the effect of repeat applications is necessary to further unlock their clear potential. The effects of bacterial extracts of *Vitreoscilla filiformis*, *Streptococcus thermophilus* and *Lactobacillus johnsonii* have yielded a promising mitigation of AD severity, likely through an antagonistic action upon *S. aureus* and *Malassezia* [231].

### 6.3. Anti-Ageing

Skin ageing corresponds with physiological changes altering elasticity, thickness and moisture. These changes are linked to multiple biological processes, including mutation accumulation, cellular senescence and microbiome dysbiosis [232]. Physiological manifestations of cutaneous ageing are accelerated by environmental factors, such as UV-mediated photodamage [233]. The composition of the skin microbiome varies throughout life, with aged skin having increased overall diversity. Ageing is further correlated to increased antimicrobial production and physiological reductions in the production of cutaneous factors, notably reduced collagen and sebum [40,234].

Correlations between ageing and cutaneous skin microbiome changes are well established. Elderly skin microbiomes are associated with consistently reduced *Cutibacterium* spp. and *Lactobacillus* spp. across multiple bodily sites. [40,235]. A conjectural association with reduced *C. acnes* abundance is reduced glycerol, fatty acids and antioxidant production perpetuating physiological changes associated with cutaneous ageing [131,236]. *Streptococcus* spp. abundance increases until puberty and correlates with younger biophysical properties, such as high elasticity [235]. The exogenous treatment of human fibroblasts with a supernatant from facial-skin-swab-derived strains of *Streptococcus* sp. resulted in an increased gene expression of collagen, filaggrin and lipid synthesis proteins [237]. The causative compound within *Streptococcus* supernatants is accepted to be spermidine, a polyamine capable of inducing cytoprotective autophagy associated with an increased turnover of cells, proteins and organelles. Spermidine is further associated with chemotherapy potentiation and tumorigenesis suppression in murine models [238,239]. Moreover, spermidine levels are shown to decrease with biological ageing, further supporting its role in ageing, and this polyamine displays low toxicity in mice and humans [240]. The supplementation of spermidine or the promotion of producing streptococci in naturally deficient older humans may provide improved phenotypes associated with biological ageing and warrant further investigation. 

Wide-ranging prospects of personal care treatments aimed to mitigate physiological alterations associated with ageing are feasible. For example, *Lactobacillus* spp. correlate with reduced photo-ageing through an ultraviolet protective effect, thus reducing collagen degradation [17]. Further, *C. acnes* RoxP is a free-radical oxygenase capable of potent antioxidant activity relevant for the mitigation of cutaneous ageing processes in purified or pre- and probiotic formulations [236,241,242].

### 6.4. Skin Rejuvenation

Topical probiotics produced with the aim of establishing stable communities of beneficial microorganisms represent a key direction for skin-microbiome-derived development. *S. epidermidis* may contribute to the production of ceramide on human skin through sphingomyelinase production leading to the digestion of antimicrobial amino alcohol sphingosines [132,243] *S. epidermidis* sphingomyelinase is sufficient to significantly increase murine ceramide content on skin and conveys no apparent pathology associated with keratinocyte cytolysis or biofilm formation [244]. Ceramides provide a diverse and vital range of functions through forming a major component of the lipid barrier to the regulation of keratinocyte proliferation and the modulation of localised immune responses [244,245]. The further potential of streptococci producing hyaluronic acid and streptococcal lysates increasing skin ceramide production highlights potential avenues that might direct skin care product development [246,247,248]. Sphingomyelinase expression appears to provide growth advantages for *S. epidermidis* by facilitating colonisation through nutrient acquisition and lipid osmoprotection [249]. Indeed, a correlative relationship was reported with reduced spingomyelinase activity in atopic dermatitis, highlighting a further direction for product development [250]. The exploration into the commercial viability of products utilising purified sphingomyelinase, lysate or probiotic formulations derived from strains with optimal activity is warranted.

*Lactobacillus* spp. were explored as topical cutaneous probiotics [225,251]. Lactobacilli are evidenced in primary human keratinocytes to aid skin rejuvenation through stimulating the expression of skin-junction proteins via Toll-like receptor 2 activation, thus promoting the integrity of the lipid barrier. Similarly, *Lactobacillus* ferment lysate stimulated keratinocyte migration leading to increased skin repair; however, the direct effector compounds are unknown [252].

### 6.5. Moisturisers

Topical moisturisers are a staple in the cosmetic industry, aiming to hydrate the skin to visually improve skin smoothness through restoration of the skin lipid barrier [253]. Since glycerol represents a potent humectant facilitating skin moisture and hydration retention, cosmetic products that aim to increase glycerol liberation from sebaceous triglycerides are a feasible option [131,254]. Hence, formulations containing either commensal-derived extracellular lipases or those that promote producer growth may yield moisturising potential. Both prebiotic and postbiotic moisturisers are on the market, with one study identifying visual improvements in skin moisture and increased skin microbiome diversity over the 4-week period studied [131,213,255]. Moisturisers containing probiotic lysates are currently commercially available containing *Lactobacillus* spp. and generally aim to stimulate the expression and production of skin barrier proteins as reported from in vivo and in vitro models [217,256,257].

### 6.6. Cutaneous Hyperpigmentation 

The superficial skin infection termed pityriasis versicolor is caused by several *Malassezia* spp. and can result in hypopigmented and hyperpigmented macules [258]. Notably, the *Malassezia* sp.-produced indole derivative compound malassezin can induce melanocyte apoptosis through aryl hydrocarbon receptor agonist activity [259,260,261,262]. This activity means malassezin was explored as a novel cosmetic treatment for facial hyperpigmentation and produces clear decreases in melanin and visible skin lightening within two to four weeks following oil–water emulsion skin treatment [261,263].

### 6.7. Rosacea

The chronic inflammatory skin disorder rosacea is characterised by persistent facial flushing, sensitive dry skin and inflamed cutaneous plaques and is associated with genetic, neurological and immune system dysregulation. Correlations between rosacea and microbial dysbiosis were identified with an overgrowth of *S. epidermidis*, *Bacillus oleronius* and reduced *C. acnes* growth [217]. However, only one known study with one participant has utilised a topical probiotic and low-dose doxycycline combinatory therapy, albeit with apparent effective results 6 months post treatment [264]; the study signposts a future scope for treatment.

## 7. Future Perspectives 

The human skin microbiome contains a wealth of compounds that can be exploited for therapeutic and personal care applications. Further research by both the public and private sectors will bring skin-microbiome-derived products to the market and importantly elucidate mechanisms of action, efficacy and safety. Simultaneously, improvements to known compounds through protein engineering that aid translation represent another promising avenue for the application of skin-microbiome-derived compounds. Such products have the capability to improve healthcare outcomes for a variety of disorders as well as improve the quality of life for a significant portion of the population.

A wealth of knowledge exists pertaining to the skin microbiome representing an incredibly well-characterised human–microbiome association. There is a plethora of commercial and therapeutic compounds evidenced to modulate the skin microbiome to provide holistic benefits. Yet many investigations of the skin microbiome are reliant on correlative studies and deductions from in vitro investigations on bacterial isolates. Hence, many microbiome investigations are only able to reveal rudimentary relationships between a limited number of species, which may not be translatable in vivo [265]. There is a necessity for the development of models more closely reflecting the natural skin environment and the diversity associated with physiological factors. Several skin models have been developed, such as stem-cell-derived reconstructed human epidermis, skin explants and NativeSkin^®^, albeit associated with a high cost and short-lived stability [266].

The advent of next-generation sequencing technologies has unequivocally transformed microbiome research. Robust metagenomic pipelines have proved pivotal for the untargeted creation of detailed compositional maps of the microbiome, facilitating procedures such as nosocomial MDR pathogen surveillance [267]. Metagenomics provides only a limited picture of the microbiome, with investigations unable to confidently identify rare species or nonviable cells [16]. Indeed, microbiome investigations are increasingly reliant on multiomics approaches facilitating revelations of nuanced microbial behaviours with potential human translatability [268].

## 8. Conclusions 

The rapid growth in the development of therapeutic and commercial skin-microbiome-derived and probiotic products illustrates the enormous potential of the field. The skin microbiome contains a vast source of cosmetic or therapeutically relevant compounds which have been shown to be translatable for the treatment of many diseases from antimicrobial-resistant pathogen infections to cancers. Many applications further exist for commercial exploitation in the personal care sector, with the potential for the development of novel cosmetic drug formulations. Similarly, the intentional modulation of the native skin flora composition through topical pre and probiotics may alleviate symptoms of many skin disorders through the promotion of microorganisms associated with healthy individuals. Skin microbiome research has recently displayed a surge in interest, yet considerable developments in areas such as in vivo modelling are required to ensure coherency and translatability to the human skin environment.

## Figures and Tables

**Figure 2 microorganisms-11-01899-f002:**
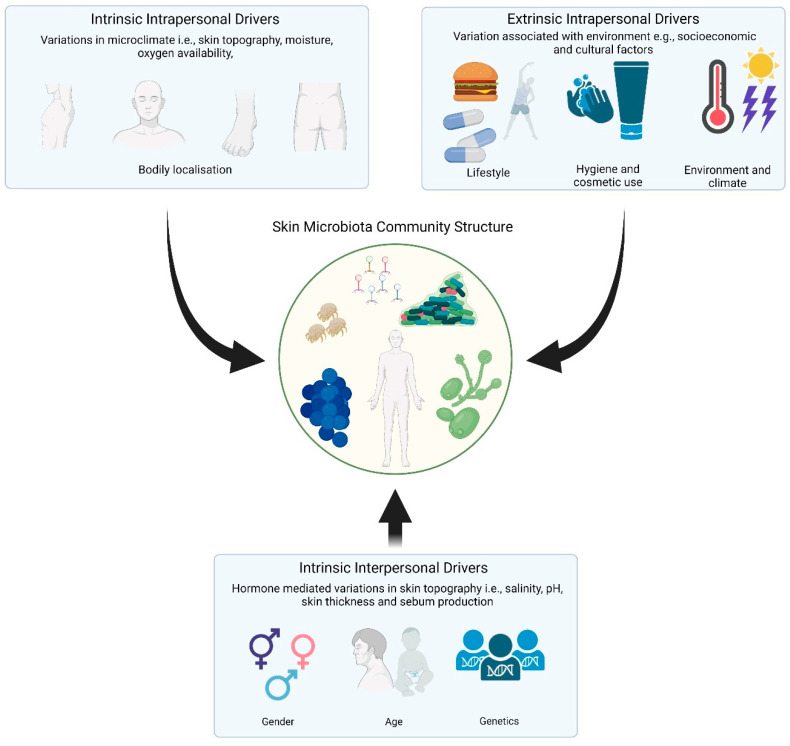
Drivers structuring the human skin microbiome structure. Intrinsic intrapersonal factors are associated with the differing physiological conditions across different microenvironments on the body. Extrinsic intrapersonal factors relate to environmentally derived variations, such as observations of cultural practices. Intrinsic interpersonal drivers are associated with the biological variation between individuals, such as hormonal differences between genders leading to altered skin physiology [45]. Created with www.BioRender.com.

## Data Availability

Not applicable.

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
