# Peer review of "Bioprospecting the Skin Microbiome: Advances in Therapeutics and Personal Care Products"

_microorganisms, 2023, doi:10.3390/microorganisms11081899_

Round 1
Reviewer 1 Report
The manuscript by Nicholas-Haizelden et al. reviewed the skin microbiome covering advances in therapeutics and advances in therapeutics. The manuscript is well written and organized. Just some minor issues:
1. Abstract is not clear. Structured and logical abstract is more attractive.
2. In the Introduction section, please describe the current research progress. How many relevant reviews have been published? What are the innovations and values of this review?
3. What are the methods and sources of literature retrieval? How to screen?
4. The literatures inserted look a little strange, such as [16]–[19].
5. A summary picture in the section of Perspectives and Conclusions could increase the readability of this paper.
Moderate editing of English language required
Author Response
The manuscript by Nicholas-Haizelden et al. reviewed the skin microbiome covering advances in therapeutics and therapeutics. The manuscript is well written and organized. Just some minor issues:
- Abstract is not clear. Structured and logical abstract is more attractive.
Response: We have revised our abstract to improve clarity and logic to, we have also maintained the format employed in by non-systematic review papers in MDPI articles.
- In the introduction section, please describe the current research progress. How many relevant reviews have been published? What are the innovations and values of this review?
Response: Thanks for raising this helpful point. We have added text (lines 69-71) that adds more context relevant to current progress and adds relevant references. to our review. Next, we add more clarity (lines 72-74) that spotlights the particular focus and novelty of our review.
- What are the methods and sources of literature retrieval? How to screen?
Response: We appreciate this remark that we did not reference the reference gathering approach used. Not being a systematic review and in keeping with the MDPI journal style we did not report a formal systematic approach.
For detail of our approach, we exhaustively searched the PubMed database using a wide variety of relevant search terms relating to the skin biology, structure and functions, topical probiotics, plus named skin diseases and disorders and personal care products. Our focus was to screen for publications within the last 10 years to add currency and then added fundamental historic papers. The four contributing authors added both academic and industry perspectives in the screen.
- The literature inserts look a little strange, such as [16]-[19]
Response: Thank you, we have modified the text accordingly.
- A summary Picture in the section of Perspectives and Conclusions could increase the readability of this paper.
Response: Thank you for this comment. We have corroborated this point of a summary picture with comments from reviewer 2 and now include a graphical abstract.
Reviewer 2 Report
The Authors present a review entitled Bioprospecting the skin microbiome: advances in therapeutics and personal care products.
The topic is definitely of interest for the scientific community because it explores a particular reality of the research and it might be helpful in moving it forward to considering the important applications that can emerge from what is described.
The manuscript is well organized and corrected in the methodology of the research, the writing is clear and concise, limits and advantages are described.
The only suggestion concerns the addition of a graphical abstract which would help the reader to immediately focus on the content of the manuscript.
I have only two minor comments:
- pagg. 5-6 line 156-7: this point should be discussed more in detail, probably it would be better to provide a short indication regarding the impact of the factors which influence the development of Healthcare Associated Infections, considering that there are the biological characteristics of the infectious agents involved as well as the susceptibility of the host to both exogenous and endogenous microorganisms.
Just some suggestions available in literature:
Beggs C, Knibbs LD, et al. Environmental contamination and hospital-acquired infection: Factors that are easily overlooked. Indoor Air. (2015) 25:462–74.
Mirande C, Bizine I, et al. Epidemiological aspects of healthcare-associated infections and microbial genomics. Eur J Clin Microbiol Infect Dis. (2018) 37:823–31.
Tozzo P, Delicati A Caenazzo L. Human microbiome and microbiota identification for preventing and controlling healthcare-associated infections: A systematic review.
Front. Public Health (2022) 10:989496.
-Pag.10 paraghraph Cancer treatments, it is better to add “Skin” (cancer treatments).
Author Response
The Authors present a review entitled Bioprospecting the skin microbiome: advances in therapeutics and personal care products.
The topic is definitely of interest for the scientific community because it explorers a particular reality of the research and it might be helpful in moving it forward to considering the important applications that can emerge from what is described.
The manuscript is well organised and corrected in the methodology of the research, the writing is clear and concise, limits and advantages are described.
The only suggestion concerns the addition of a graphical abstract which would help the reader to immediately focus on the content of the manuscript.
Response: Thanks for your comments. The point about a graphical abstract agrees with reviewer 1 and the need for a focus image. We have generated a graphical abstract that to support our article.
I have only two minor comments:
- 5-6 line 156-7: this point should be discussed in more detail, probably it would be better to provide a short indication regarding the impact of the factors which influence the development of Healthcare Associated Infections, considering that there are the biological characteristics of the infectious agents involved as well as the susceptibility of the host to both exogenous and endogenous microorganisms.
Just some suggestions available in the literature:
Environmental contamination and hospital-acquired infection: factors that are easily overlooked
Epidemiological aspects of healthcare-associated infections and microbial genomics
Human microbiome and microbiota identification for preventing and controlling healthcare-associated infections.
Response: Thank you for your comments. We have now included mention of hospital-acquired infections and the relevance of bacteriocin treatments and added references you provide (lines 148-151)
- 1 paragraphs Cancer treatments, it is better to add “Skin” (cancer treatments).
Response: Thank you for this comment - this change adds clarity to the section.